# Transcriptome and metabolome analyses reveal regulatory networks associated with nutrition synthesis in sorghum seeds
Adil Khan[1,4], Ran Tian[1,4], Scott R. Bean[2], Melinda Yerka[3] & Yinping Jiao [1] ✉

Cereal seeds are vital for food, feed, and agricultural sustainability because they store and provide essential nutrients to human and animal food and feed systems. Unraveling molecular processes in seed development is crucial for enhancing cereal grain yield and quality. We analyze spatiotemporal transcriptome and metabolome profiles during sorghum seed development in the inbred line 'BTx623'. Morphological and molecular analyses identify the key stages of seed maturation, specifying starch biosynthesis onset at 5 days post-anthesis (dpa) and protein at 10 dpa. Transcriptome profiling from 1 to 25 dpa reveal dynamic gene expression pathways, shifting from cellular growth and embryo development (1–5 dpa) to cell division, fatty acid biosynthesis (5–25 dpa), and seed storage compounds synthesis in the endosperm (5–25 dpa). Network analysis identifies 361 and 207 hub genes linked to starch and protein synthesis in the endosperm, respectively, which will help breeders enhance sorghum grain quality. The availability of this data in the sorghum reference genome line establishes a baseline for future studies as new pangenomes emerge, which will consider copy number and presence-absence variation in functional food traits.

Sorghum [*Sorghum bicolor* (L.) Moench] stands out as a versatile and climate-smart crop, ranking among the world's top five cereals in terms of production. It plays a crucial role in providing dietary calories and essential nutrients for a substantial proportion of the global population[1–4]. The challenges posed by population growth, climate change, and the increasing demand for nutritious cereal crops underscore the need to enhance both the quantity and quality of sorghum grain production[5–10]. To meet these challenges, plant breeders need a comprehensive understanding of the molecular, biochemical, and physiological mechanisms governing sorghum seed development. Such insights will ensure an ample and nutritious food supply in the face of climate change.

The sorghum seed is a complex system comprised of genetically distinct tissues: a diploid embryo, a triploid endosperm, and diploid maternal tissues[11–13]. Following double fertilization, an evolutionarily conserved process in all flowering plants, the zygote develops into the embryo while the central cell transforms into the endosperm. The endosperm serves as a nutrient-rich storage tissue, supplying the energy required for the initial growth of the embryo and subsequent germination in monocots[11,14,15]. The developmental timeline from fertilization of the ovule to seed maturity in sorghum is typically 40–45 days[16]. Initially, from 3–5 days post-anthesis

(dpa), there is limited growth, and no apparent development of the embryo or endosperm[11,17]. Subsequently, endoreduplication occurs, followed by starch accumulation. While starch accumulation in maize initiates at 10 dpa[18,19], in sorghum, it commences at 5 dpa[11]. From 6–24 dpa, the caryopsis, embryo, and endosperm undergo rapid growth, accompanied by significant changes in seed size. However, from 24–35 dpa, the growth rate diminishes, and only slight alterations in the sizes of the caryopsis, embryo, and endosperm occur[16]. These observations indicate three primary developmental stages in the sorghum caryopsis: an early stage before 6 dpa, a middle stage spanning 6–24 dpa, and a late stage extending from 25–35 dpa[20].

Starch metabolism is a dynamic physiological process that is required for energy storage and utilization[21]. It involves a sophisticated interplay between sucrose metabolism and various tightly regulated pathways governed by many enzymes[22]. Among those enzymes are ADP-glucose pyrophosphorylase (AGPase) and various starch synthases (SSs), starch branching enzymes (SBEs), and starch debranching enzymes (DBEs). In sorghum, kafirins are the predominant seed storage proteins, constituting 77 to 82% of endosperm protein and 68 to 73% of total protein in whole sorghum grain[23,24]. Non-prolamin proteins, namely albumins, globulins,

[1]Institute of Genomics for Crop Abiotic Stress Tolerance, Department of Plant and Soil Science, Texas Tech University, Lubbock, TX 79409, USA. [2]Grain Quality and Structure Research Unit, Center for Grain and Animal Health Research, USDA-ARS, 1515 College Ave, Manhattan, KS 66502, USA. [3]Department of Agriculture, Veterinary & Rangeland Sciences, University of Nevada-Reno, Reno, NV 89557, USA. [4]These authors contributed equally: Adil Khan, Ran Tian. ✉e-mail: yijiao@ttu.edu

and glutelins, make up the remaining 20% of protein. Kafirins have molecular weight-based classifications as α-kafirins (25–23 kDa), β-kafirins (20–16 kDa), γ-kafirins (28–50 kDa), and δ-kafirins (13 kDa)[25–28]. A total of 27 previously reported kafirin genes in the sorghum genome include 23 α-kafirins, 1 β-kafirin, 2 γ-kafirins, and 1 δ-kafirin[24]. Kafirins exist as monomeric proteins, small oligomeric protein complexes, and large polymeric protein complexes, held together by inter-protein disulfide bonds. Sorghum kafirins can be further classified as kafirin 1 and 2 based on solubility during protein extraction. Kafirin 1 comprises proteins not heavily cross-linked into large polymeric structures, while kafirin 2 is solubilized from the remaining large polymeric complexes[29]. The ratio of kafirin 1 to kafirin 2 is a crude measure of protein cross-linking in the sorghum seed[30].

Seed development is a process that is notable for dynamic physiological and biochemical changes[31,32]. The chemical composition of mature seeds is shaped by complex gene expression networks. Recent years have seen a surge in transcriptomic analyses investigating seed development in diverse plant species, including *Arabidopsis thaliana*, *Oryza sativa*, *Triticum*, *Zea mays*, *Paeonia*, *Medicago truncatula*, *Brassica napus*, *Hordeum vulgare*, and *Glycine max*[33–59]. These studies have advanced our understanding of seed spatiotemporal gene expression patterns and their regulation, offering genetic insights that are applicable to breeding for quality traits. For instance, high-throughput RNA sequencing (RNA–Seq) in maize identified genes and transcription factors (TFs) strongly associated with amylose and amylopectin biosynthesis[60]. Similarly, transcriptome analyses of developing soybean seeds revealed hub genes implicated in oil and protein accumulation[55,61]. The integrated metabolomic and transcriptomic analyses of rice seeds identified candidate genes involved in the structural modification of anthocyanins[62]. These examples underscore the potential for integrating metabolomic and transcriptomic information during seed development to clarify the molecular mechanisms driving the accumulation of desirable chemical profiles. Consequently, integrating the transcriptomic and metabolomic profiles of developing sorghum seeds holds promise for generating new molecular breeding resources, thereby enhancing sorghum seed quality and yield for a hungry planet.

Despite the importance of sorghum in global food and feed systems, a significant gap remains in our understanding of gene expression dynamics during sorghum endosperm and embryo development. A recent study reported the transcriptome of developing sorghum seeds at various timepoints from 5 to 25 dpa, providing insights into overall seed development, but the transcriptomes of the embryo and endosperm were not differentiated[63]. Additionally, there are no published studies that comprehensively explore the transcriptomic and metabolomic networks governing carbon allocation tradeoffs driving the accumulation of starch and protein throughout seed development, which is a major target for plant breeding. Research is particularly needed to clarify stage- and tissue-specific gene expression and crosstalk within and among the embryo, endosperm, and whole seed[64–66].

To address this knowledge gap, we conducted an in-depth transcriptomic analysis of developing sorghum seeds, dissecting the early whole seed, embryo, and endosperm tissues from fertilization through maturity. Complementing these efforts, metabolomic analyses were performed at five key seed developmental stages to gain insights into the accumulation of specific metabolites that drive nutritional quality, ultimately contributing to the mature grain quality profile. Our findings have unveiled hub genes and metabolites crucial for regulating mature sorghum seed chemistry profiles, broken down by their specificity to the embryo, endosperm, and/or the whole seed. This comprehensive analysis marks a significant step toward unraveling the intricate molecular mechanisms underlying sorghum seed development, with implications for enhancing nutritional quality and overall yield.

## Results
### Morphological analyses of sorghum seed development
To comprehensively characterize sorghum seed development, we collected developing seed samples from the reference genome line 'BTx623' spanning 1–25 dpa (Supplementary Data 1). Over this period, the seed coat exhibited a transition from bright green (1–12 dpa) to light green (13–21 dpa), ultimately a yellowish-green hue (22–25 dpa) (Fig. 1a). The fresh weight of the seeds, depicted in Supplementary Fig. 1a, exhibited a gradual increase post-pollination, reaching its peak at 22 dpa (average 1.23 g/50 grain). Notably, the rate of seed weight gain during early stages (1–15 dpa) surpassed that of later stages (16–25 dpa). By the final timepoint of the study at 25 dpa, sorghum seeds had entered the desiccation stage, making the separation of the embryo and endosperm challenging and justifying the termination of the experiment.

Scanning electron microscopy (SEM) imaging revealed the presence of starch granules at 5 dpa (Fig. 1b), with the quantity and dimensions of these granules gradually increasing in subsequent developmental phases. These observations align with previous research indicating that endoreduplication precedes starch accumulation in sorghum[11]. Concurrently, kafirin 1 (monomeric proteins and small oligomeric complexes) and kafirin 2 (polymeric cross-linked complexes) emerged at low levels between 5 and 10 dpa but increased between 15 and 25 dpa. This suggests a crucial developmental shift in kafirin accumulation and crosslinking between 10 and 15 dpa. The abundance of kafirin 1 surpassed that of kafirin 2 at all stages (Fig. 1c). At 25 dpa, kafirin 1 constituted approximately 84% of the total protein content, while kafirin 2 comprised the remaining 16% (Fig. 1c). Consequently, the sampled timepoints in this study (5, 10, 15, 20, and 25 dpa) were representative stages of sorghum seed development based on SEM.

### Dynamic metabolic changes during sorghum seed development
Differential metabolite accumulation was assessed across the five sampled timepoints (Supplementary Data 2; Fig. 2a). Among 7959 detected peaks, a total of 2073 metabolites were successfully identified and 955 were assigned to functionally annotated pathways (Supplementary Fig. 1b). The functionally annotated metabolites were grouped into 13 functional categories (Supplementary Fig. 1c; Supplementary Data 3). The top enriched pathways within these categories included the biosynthesis of secondary metabolites (20.23%), amino acid metabolism (18.75%), lipid metabolism (13.98%), and carbohydrate metabolism (13.09%) (Supplementary Fig. 1d). Pathway analysis of the six clusters of metabolites, reflecting different stages of development (Supplementary Fig. 2a), revealed that starch biosynthesis, initiates at 5 dpa, transitioning to protein biosynthesis and degradation after 15 dpa (Supplementary Fig. 2b–g). This observation agreed with the SEM imaging of starch granules and kafirin quantification (Fig. 1b, c). Taken together, these results suggest that the starch and fatty acid contents in sorghum seeds were determined before final protein content, potentially contributing to the well-known negative correlation between starch and protein content.

Principal component analysis (PCA) highlighted distinct variations in metabolite profiles across the five timepoints (Supplementary Fig. 3; Fig. 2a). A total of 1495 compounds exhibited differential accumulation throughout seed development (Fig. 2b; Supplementary Data 4), indicating a stage-specific accumulation pattern. Up-regulated metabolites between 10 and 5 dpa were associated with the biosynthesis of fatty acids, linoleic acid, sugar metabolism, and lysine, while down-regulated metabolites were linked to flavanol biosynthesis and the pentose phosphate pathway (Fig. 2c). This pattern suggested a resource reallocation favoring essential processes during the early stages of seed development, with upregulation of high-energy molecules and downregulation of metabolites related to secondary metabolism and nucleotide synthesis. Similar trends were observed in comparisons between the 15 and 5 dpa (Fig. 2d). In contrast, comparisons between the 20 and 5 dpa, as well as the 25 and 5 dpa, revealed a shift toward alanine, aspartate, glutamate, flavonoid, and linoleic acid biosynthesis, indicating an emphasis on protein synthesis during later stages of seed development (Fig. 2e & f). The 189 metabolites that were consistently up-regulated and the 234 consistently down-regulated metabolites across the five timepoints during seed development (Supplementary Fig. 4a–c) likely play roles in the accumulation of sorghum protein, starch, and oil. The consistently up-

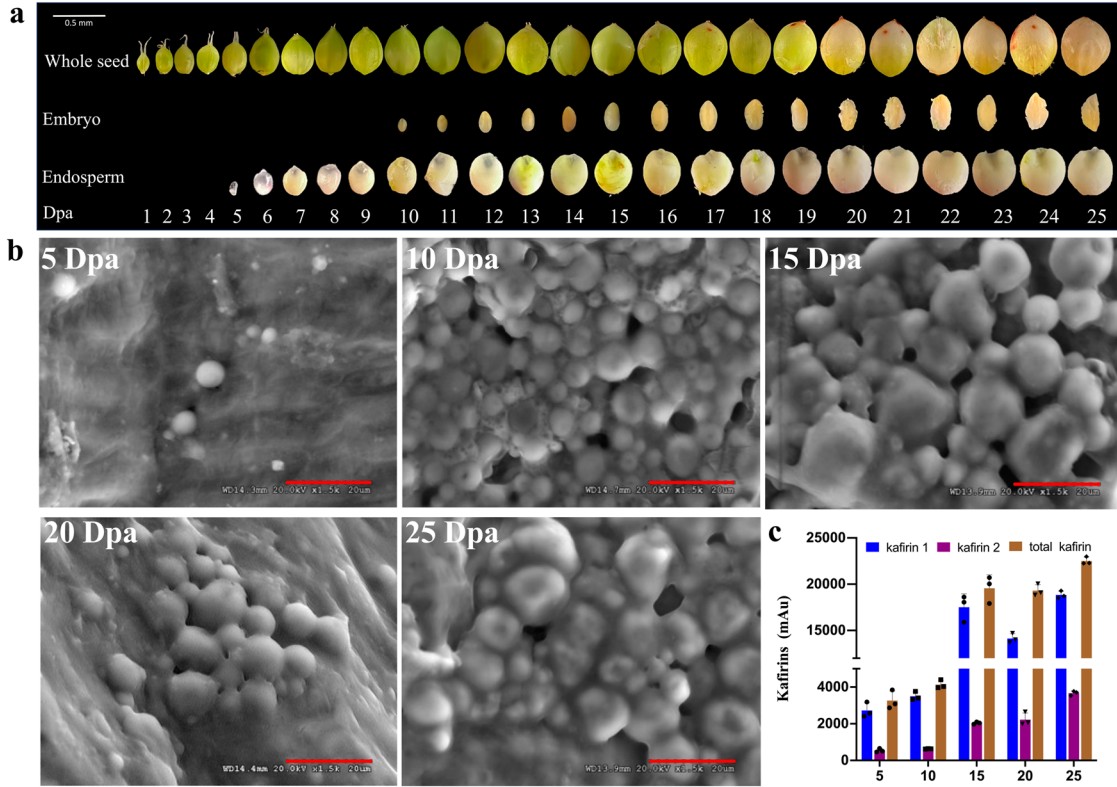

**Fig. 1 | Morphological changes, structural alterations, and kafirin accumulation during sorghum seed development. a** Sequential morphological transitions of BTx623 sorghum whole seed, embryo, and endosperm at various developmental stages. **b** Cryo-scanning electron microscopy (cryo-SEM) images illustrating the ultrastructural changes in BTx623 sorghum endosperms at 5, 10, 15, 20, and 25 dpa. Bar: 20 μm. **c** Quantification of kafirin 1, kafirin 2, and total protein content in BTx623 sorghum seeds at 5, 10, 15, 20, and 25 dpa. Error bars denote the standard error of three replicates. *n* = 3.

regulated metabolites were associated with the synthesis of lipids, phenolic acids, and flavonoids (Supplementary Fig. 4d), while the down-regulated metabolites were enriched in linoleic acid, monoterpenoid biosynthesis, and the biosynthesis of unsaturated fatty acids (Supplementary Fig. 4e). Collectively, these trends suggested that lipid metabolism, secondary metabolite production, and nucleotide biosynthesis undergo dynamic modulation throughout seed development.

**The transcriptome landscape of sorghum seed development**
Transcriptome profiling of sorghum seed development encompassed 45 samples (1–9 dpa for the early whole seed, 6–25 dpa for the endosperm, and 10–25 dpa for the embryo), each with two replicates (Supplementary Data 5 & 6). We obtained over 218.8 million high-quality reads, averaging 23.78 million reads per replicate (Supplementary Data 5). The robust correlation (average $R^2$ = 0.976) between the two replicates for each sample underscored the high quality of the data (Supplementary Data 5). Additionally, the qPCR results from three replicates of four randomly selected genes at the five major timepoints (5, 10, 15, 20, 25 dpa) closely matched the RNAseq data (Supplementary Fig. 5), further validating the reliability of our findings.

A total of 21,971 genes were expressed (FPKM ≥ 1) during sorghum seed development (Supplementary Data 7). More genes were expressed in early whole seeds and endosperms than in later stages (Supplementary Fig. 6a), indicating heightened gene activity during early seed development as tissue types and specialized cell layers initially diversify. The higher expression (average FPKM of all genes from 10–25 dpa) in the embryo compared to the endosperm suggests greater metabolic activity and more complex developmental processes in the embryo. The distinctiveness of the transcriptome landscapes between these tissues was further confirmed by the greater median gene expression level in the embryo compared to the endosperm (Supplementary Fig. 6b).

Among the 2049 genes specifically expressed in the 1–9 dpa whole seed (Supplementary Fig. 6c), 1558 were previously identified in the sorghum ovary cell wall using RNA-Seq data[67]. This was expected, as whole sorghum seeds (caryopses) are comprised of distinct maternal (pericarp, derived from the ovary cell wall) and daughter (embryo, endosperm) tissues (Supplementary Fig. 6d). Similarly, 795 embryo-specific genes were enriched for pathways related to embryogenesis (Supplementary Fig. 6e), while 397 endosperm-specific genes were enriched in metabolic and mitogen-activated protein kinase (MAPK) signaling pathways (Supplementary Fig. 6f).

A PCA of the transcriptome data effectively separated developing seeds into three groups based on their tissue identity, validating their distinct developmental activities (Fig. 3a). Early whole seed samples collected at 1–5 dpa formed a separate cluster from samples collected at 6–9 dpa. The latter cluster exhibited proximity to the endosperm sample, indicating shared gene activity within the whole seed and young endosperm. In the hierarchical clustering of gene expression within the embryo, the first and second clusters were associated with morphogenesis and maturation processes, respectively (Fig. 3b). This aligns with the embryo's sequence of active DNA synthesis, cell division, and differentiation in early and middle phases, followed by the synthesis of storage reserves and desiccation processes in later phases[17,20,46,68]. The three endosperm clusters aligned with canonical stages guiding harvest times, encompassing the milk, soft dough, and hard dough phases (Fig. 3c). The milk phase coincided with cellularization, following the syncytial phase which involves the formation of cell walls and the partitioning of the endosperm into discrete cells. The soft dough and hard dough phases correlated with the grain-filling phase, characterized by the development of distinct cell types and the accumulation of storage reserves[17,20].

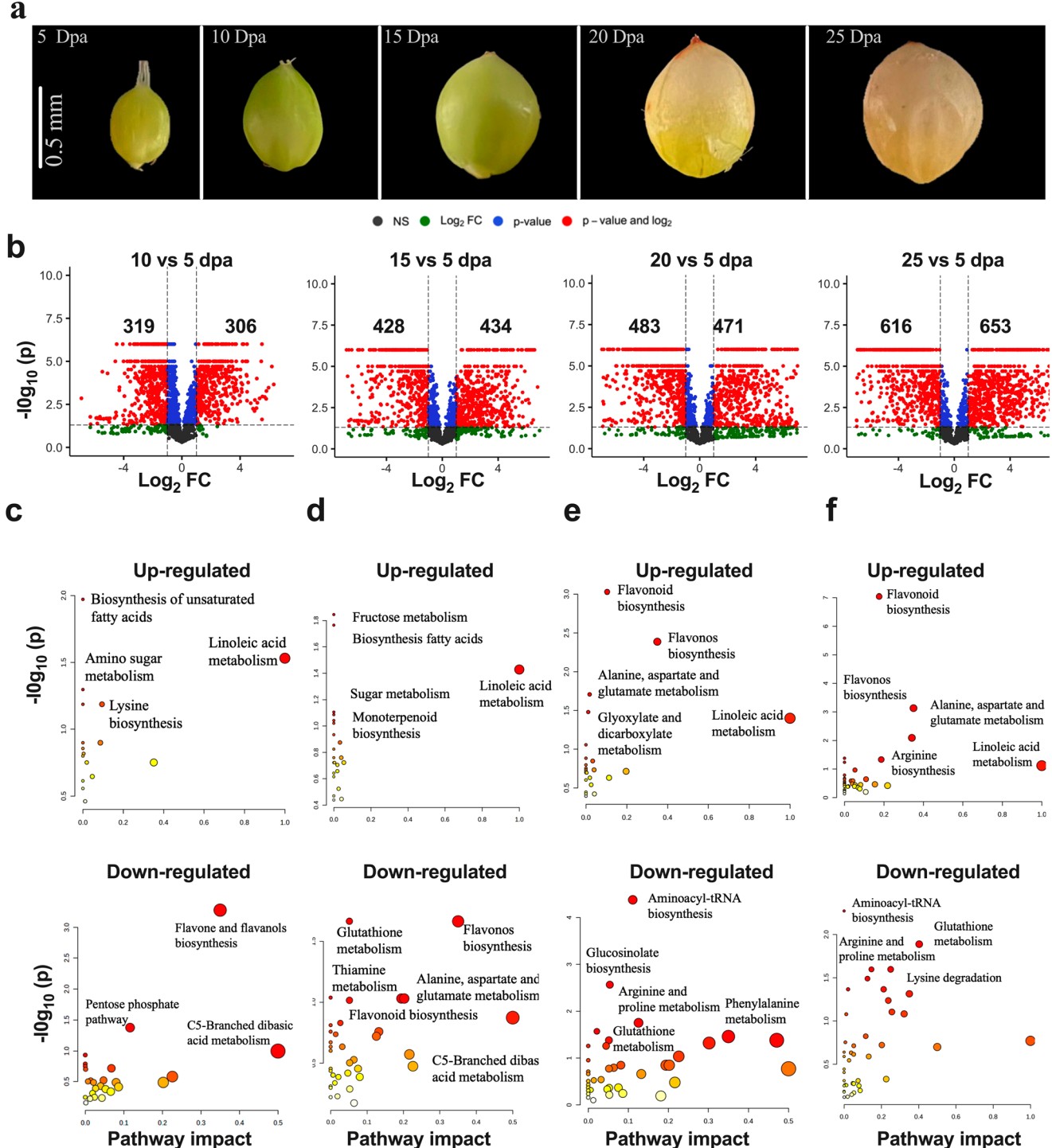

**Fig. 2 | Overview of differentially expressed metabolites identified through pairwise comparisons at key timepoints during sorghum seed development.** **a** Morphological transitions in BTx623 sorghum seeds at five critical timepoints (5, 10, 15, 20, and 25 dpa) subjected to metabolome analysis. **b** Volcano plot displaying metabolite features, with numbers indicating significantly upregulated [$\log_2$ (fold-changes) ≥ 1.5; $q$-values ≤ 0.05] and downregulated [$\log_2$ (fold-changes) ≤ -1.5; $q$-values ≤ 0.05] metabolites in the specified comparisons. Gray-dashed lines represent the q-value and fold-change filter. **c–f** Categorization of upregulated and downregulated metabolic pathways according to the Kyoto Encyclopedia of Genes and Genomes (KEGG) for different comparison groups (10 Vs. 5 dpa, 15 Vs. 5 dpa, 20 Vs. 5 dpa, and 25 Vs. 5 dpa). Node colors indicate $p$-values, with white and red denoting lower and higher $p$-values, respectively. Node radii correspond to pathway impact values, with smaller and larger radii indicating lower and higher impact values, respectively.

## Main pathways involved in sorghum seed development

To identify the active cellular processes in developing sorghum seeds, we employed a $k$-means clustering methodology, which revealed 12, 16, and 15 co-expression clusters in the early whole seed, embryo, and endosperm, respectively (Fig. 4a, b; Supplementary Fig. 7; Supplementary Data 6). In early whole seeds (1–9 dpa), clusters c1–c6 (1–5 dpa; early stage) were enriched in genes controlling cellular growth, proliferation, and fundamental structures essential for seed development (Supplementary Fig. 7). In contrast, clusters c7–c11 (5–9 dpa; middle stage) were enriched in processes related to embryo development and storage compound accumulation like

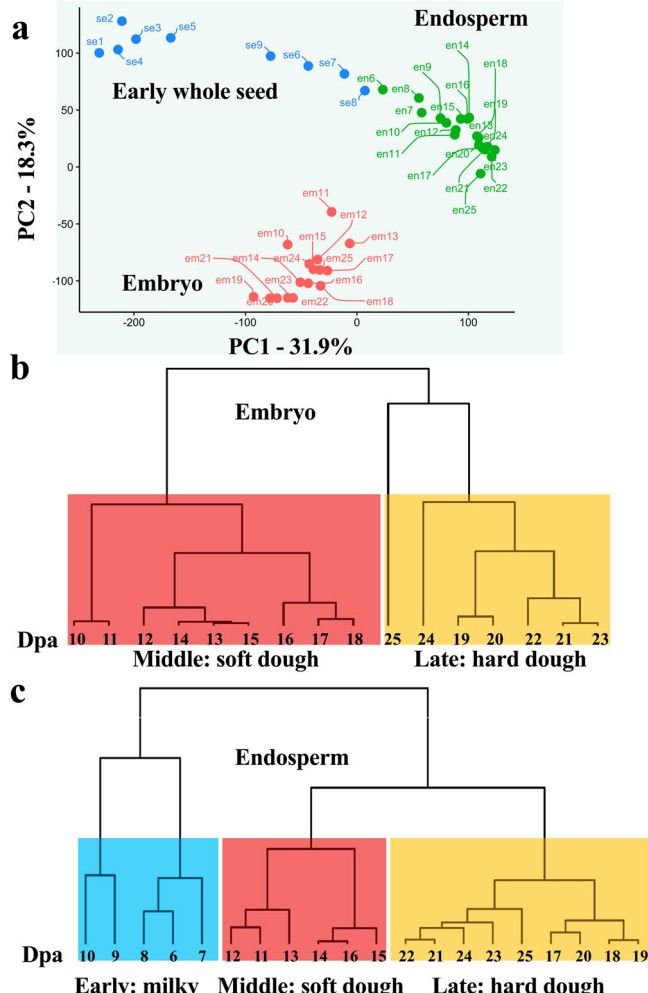

**Fig. 3 | Global transcriptome relationships among different developmental stages and tissues. a** Principal Component Analysis of the RNA-seq data for the 45 seed samples (1–9 dpa of the early whole seed, 6–25 dpa of the endosperm, and 10–25 dpa of the embryo) revealing three distinct clusters corresponding to each tissue. Cluster dendrogram displaying the global transcriptome relationships among time series samples of the embryo (**b**) and endosperm (**c**). The lower row indicates the developmental phases, as per the cluster dendrogram of the time series data, with numbers indicating days post-anthesis.

starch biosynthesis, which aligned with SEM imaging showing starch granules emerging at 5 dpa. Genes constitutively expressed from 1–9 dpa (c12) were associated with endosomal vesicle fusion, vacuolar acidification, Nicotinamide Adenine Dinucleotide (NAD) biosynthesis, organelle fusion, vacuole organization, and embryo development.

In the embryo samples, primary active pathways were related to cell division, fatty acid biosynthesis, and embryo development. The middle stage (c1–c7) of embryo development (10–18 dpa) was enrichment in pathways regulating cellular processes, including the cell cycle, starch, and amino acid biosynthesis. The later stage (c8–c15; 19–25 dpa) indicated active embryo growth with enriched fatty acid and lipid biosynthesis pathways, suggesting a crucial role in providing energy and nutrition. Genes in c16, expressed across all stages, were enriched for embryo development, protein folding, membrane organization, and transport, indicating their fundamental roles across all timepoints (Fig. 4a).

In the endosperm, a distinct shift toward the activation of storage pathways (starch and storage proteins) occurred after initial cell division in early stages of development (Fig. 4b). Clusters c1–c7 (6–14 dpa) included genes related to cell cycle processes, metabolic processes, and seed storage compound biosynthesis, indicating their roles in regulating nutrient storage

and energy metabolism. During middle and late stages, extensive growth and differentiation in the endosperm coincided with the accumulation of storage reserves for starch and protein. Genes in c15, expressed throughout endosperm development, highlighted an emphasis on cellular metabolism and function. Programed cell death (PCD) is a crucial process for the cereal endosperm as it transitions from cell division to nutrient accumulation[69,70]. Ethylene is one of the major hormones involved in the PCD[71]. Based on the ethylene biosynthesis enzymes in rice[72], we noted the peak expression of most of these genes during the early stages of sorghum endosperm development (6–10DPA, Supplementary Fig. 8a). Notably, the primary enzyme initiating ethylene synthesis methionine adenosyltransferase (SAM), encoded by the *Sobic.003G151600* and *Sobic.009G033600* genes, exhibited a pronounced downregulation trend during sorghum endosperm development. This observation suggests that ethylene acts as a negative regulator of grain filling and PCD. Consistently, other documented PCD regulator genes in maize, such as *ZmDEK40*[73], *ZmDEK66*[74], *ZmATR*, and *ZmATM*[75], showed similar downregulation trend in the process of sorghum endosperm development (Supplementary Fig. 8b).

## Hub genes and key networks associated with starch and protein synthesis

Understanding the regulation of biosynthetic networks that govern seed nutrition is crucial for enhancing sorghum grain quality. The presented data highlighted that starch and protein biosynthesis in BTx623 sorghum seeds predominantly took place in the endosperm (Supplementary Fig. 9a, b; Supplementary Data 8), aligning with its role as a storage tissue for energy during germination and early seedling growth[76].

In endosperm, the expression patterns of sorghum ortholog genes associated with starch and kafirin biosynthesis revealed distinct trends (Supplementary Fig. 9c, d). Starch biosynthesis was most active between 5–15 dpa, while protein biosynthesis primarily occurred during 15–25 dpa. Interestingly, kafirin genes constituted 44.77% of total endosperm transcripts from 6–25 dpa, with a notable increase from 24.67% (6–15 dpa) to 62.16% (16–25 dpa), indicating predominant kafirin synthesis post-15 dpa (Supplementary Fig. 10a). The most abundant kafirin gene transcripts during endosperm development were α-kafirins (34.20%), followed by γ-kafirins (6.99%), β-kafirins (3.29%), and δ-kafirins (0.277%) (Supplementary Fig. 10a). This agrees with the first observation of starch granules at 5 dpa in the SEM imaging and the distinct metabolite enrichments at the five major timepoints. Throughout all stages, the average expression level of kafirin and starch synthesis genes was higher in the endosperm than in the embryo (Supplementary Fig. 10b). For example, a significant proportion of kafirin genes (23 out of 27) ranked among the 100 most highly expressed genes in the endosperm, compared to only 9 out of 27 genes in the embryo (Supplementary Fig. 10c; Supplementary Data 9 & 10).

A co-expression network analysis using the 20,491 genes expressed in the endosperm (FPKM ≥ 1) was conducted to scrutinize the regulation of starch and protein biosynthesis. Soft clustering was employed to allow genes to belong to multiple clusters. Among the 12 co-expression modules (Supplementary Fig. 11; Supplementary Data 11), modules 8 and 12 exhibited significant enrichment (FDR < 0.05) in starch biosynthesis-related genes, as determined by Fisher's Exact Test (Fig. 5a, Supplementary Fig. 12 a, b). In addition, genes from the same modules were associated with diverse functional categories such as proteosome activity, N-glycan biosynthesis, participation in the tricarboxylic acid (TCA) cycle, spliceosome activity, amino acid synthesis, oxidative phosphorylation, and DNA replication (Fig. 5b). Gene Network Analyzer analysis of these modules identified 361 as hub genes based on two criteria: gene degree of connectivity ≥ 5 in the hub module and gene module membership > 0.8. Many hub genes encode proteins participating in the TCA cycle, ribosome biogenesis, oxidative phosphorylation, DNA replication, starch, and sucrose metabolism. For example, the top 10 highly connected genes (Supplementary Data 12) included genes that code for succinate dehydrogenase, metallopeptidase M24 family proteins, the translation initiation factor 3B family, and elongation factor 1-gamma 3. These results indicate that the core enzymes in

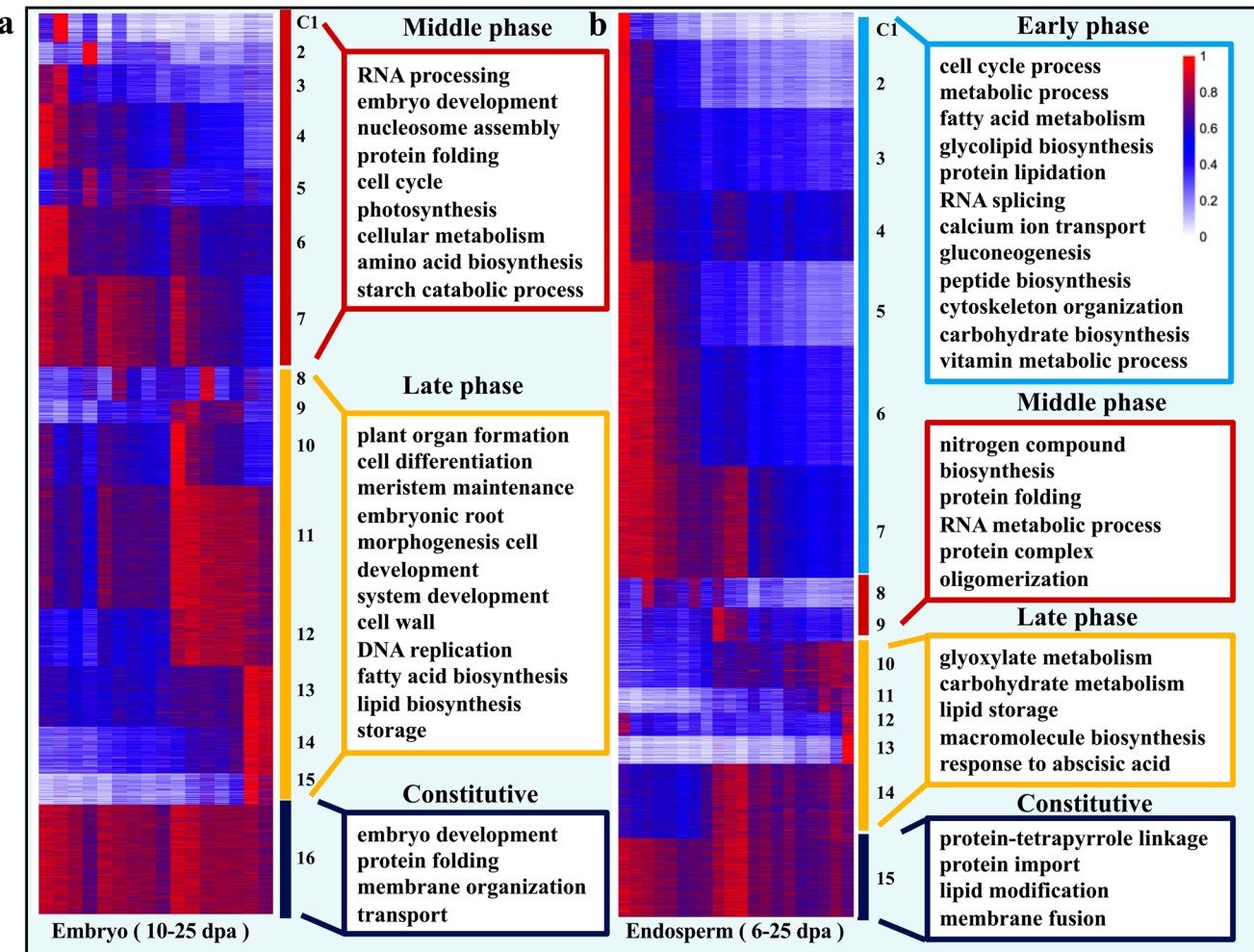

**Fig. 4 | Gene expression patterns and functional transitions over the time course for the BTx623 sorghum embryo and endosperm. a** Expression patterns of co-expression clusters for the embryo. **b** Expression patterns of co-expression clusters for the endosperm. These patterns are organized according to the sample timepoints at which they peak. Functional categories enriched within different co-expression clusters for the embryo and endosperm are listed, reflecting various stages of sorghum seed development. For each gene, the RPKM value is displayed, normalized in relation to the maximum RPKM value observed for that gene across all timepoints.

starch synthesis are regulated by the identified hub genes. For instance, *Sobic.007G023400*, one of the hub genes, encodes the succinate dehydrogenase iron-protein subunit (SDHB), a crucial component of the succinate dehydrogenase enzyme complex that is essential for the TCA cycle, Krebs cycle, and the electron transport chain during cellular respiration[77]. Notably, two genes involved in starch branching, *Sobic.001G083900 (SbPHOL)* and *Sobic.003G358600 (SbPHOH)*, were also identified as hub genes, emphasizing their regulatory role in starch biosynthesis.

Among the 12 co-expression modules in the endosperm (Supplementary Fig. 11), modules 10 and 4 exhibited significant enrichment (FDR < 0.05) for kafirin genes (Fig. 5c; Supplementary Fig. 12c). Specifically, Module 4 encompasses 15 α-kafirin genes, while Module 10 includes six α-kafirin genes and two γ-kafirins. The β-kafirin and δ-kafirin genes were present in Modules 2 and 5, respectively. These findings indicated that different types of kafirins are synthesized at different stages of seed development. Notably, β- and δ-kafirins were expressed exclusively in the later stages of endosperm development (20–25 dpa), while α-kafirin expression spans from 15 to 25 dpa in wild-type sorghum.

The 1719 genes within modules 4 and 10, including 23 kafirin genes, were functionally implicated in crucial biological processes such as carbon metabolism, lipid metabolism, the MAPK signaling pathway, and seed storage protein processes, based on GO term analysis (Fig. 5d). The function of genes co-expressed with seed storage proteins could imply their involvement in the biosynthesis, accumulation, and/or mobilization of these

proteins during seed development. Subsequently, we identified 207 hub genes related to kafirin biosynthesis from modules 4 and 10 (Supplementary Fig. 12d; Supplementary Data 13). These genes were significantly enriched in various biological processes such as lipid metabolism, fatty acid degradation, amino acid biosynthesis and degradation, MAPK signaling, carotenoid biosynthesis, and hormonal signaling. The top 10 most highly connected genes are presented in Supplementary Data 13. Some of these top hub genes have been investigated for roles in protein synthesis in other crops. For instance, extra-large GTP-binding proteins have been reported to play key roles in regulating panicle architecture, plant growth, development, grain weight, and disease resistance[78]. Similarly, bZIP TF is known to play a key role in regulating various biological pathways, including seed storage protein biosynthesis[79–82]. Further characterization of various alleles of these genes could provide valuable insights into the molecular mechanisms and regulatory networks underlying these processes and indicate gene targets for modifying protein content through molecular breeding.

## Discussion

Sorghum is a major climate-smart cereal crop that will continue to play a significant role in adapting global food and feed systems to climate change. However, the molecular mechanisms governing sorghum seed development have not yet been explored as extensively as in other cereals[44,45,83–88]. Previous characterization of the transcriptomic landscapes of endosperm development in wheat, rice, oat, barley, and maize have provided a solid foundation

**Fig. 5 | Identification of hub genes associated with starch and kafirin biosynthesis.** Clustering analysis using the Mfuzz package. The fuzzy c-means clustering algorithm uses a soft partitioning clustering method. Twelve co-expression modules were obtained. Yellow or green lines correspond to genes with a low membership value, whereas red and purple lines correspond to genes with a high membership value. Most genes showed a high membership value. Module numbers and the corresponding *P*-value are included above each cluster. **a** Modules 8 and 12 showed significant enrichment (FDR < 0.05) for starch biosynthesis-related genes using Fisher's Exact Test. **b** GO-enrichment analysis of genes enriched in modules 8 and 12, visualized as a network. **c** Modules 8 and 12 showed enrichment (FDR < 0.05) for kafirin biosynthesis-related genes using Fisher's Exact Test. **d** GO-enrichment analysis of genes enriched in modules 4 and 10, visualized as a network. Node colors represent enrichment significance (FDR < 0.05), node size indicates gene set size, and edge thickness signifies gene overlap. The analysis was performed using ShinyGO.

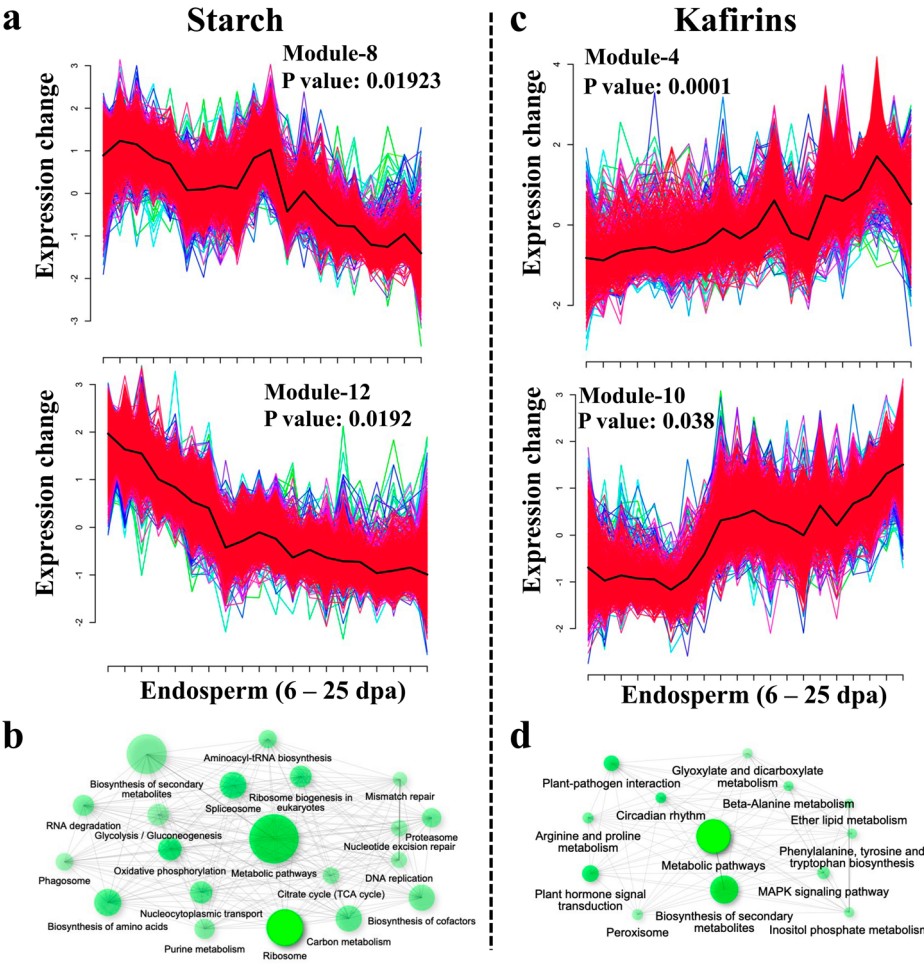

for the present study. Here, we employed RNA-Seq and untargeted metabolome profiling to capture dynamic transcriptome and metabolome profiles of sorghum seed development. Our findings provide significant insights into the interplay of gene expression and metabolite accumulation across five developmental timepoints, offering a comprehensive temporal perspective on functional and cellular specialization in the whole seed (maternal + daughter tissues), embryo, and endosperm (daughter tissues).

Our research has shed light on the dynamic landscape of transcriptomic and metabolomic activity during sorghum seed development, focusing on the reference genome line BTx623. This study lays the groundwork for clarifying the significant diversity observed in seed traits within global sorghum mapping and breeding populations. For instance, a genome-wide association study involving 837 varieties revealed 81 quantitative trait loci (QTLs) associated with grain size, which influences starch and protein harvestable yields on a per-area basis[89]. Other studies have documented the extensive diversity in grain quality traits across sorghum germplasm, including variations in seed color, starch, protein, and oil contents[90–92]. This diversity underscores the multitude of phenotypic variants impacted by sorghum seed development.

Future investigations are needed in diverse sorghum varieties to elucidate how genetic variation in starch, protein, and oil biosynthetic pathways results in differential carbon partitioning among them; and how that partitioning is impacted by whole-plant phenotypes and local adaptation. Comparative genomics studies should focus on differences in seed development processes among varieties having genetic variation in hub genes to shed light on tissue-specific metabolite accumulation patterns and contribute to improved seed quality, stress responses, and adaptation to local production environments. Hence, the baseline information presented herein about sorghum seed developmental programing holds promise for

enhancing the utility of sorghum in breeding for climate-smart food and feed systems.

Tissue-specific (TS) genes play a pivotal role in unraveling the mechanisms governing tissue or organ identity and can be crucial in guiding the progression through seed developmental programs[93,94]. In our study, we identified 499 TS genes, including 41 TFs, expressed specifically in early whole seeds, embryos, and endosperms (Supplementary Data 6; Supplementary Fig. 13a). Analyses revealed variations in the numbers of TS genes among embryos (127 genes, including 14 TFs), endosperms (71 genes, including 6 TFs), and early whole seeds (79 genes, including 12 TFs), with the endosperm exhibiting the lowest number of TS genes (Supplementary Fig. 13a), consistent with previous findings in maize[95] and wheat[88]. This may suggest the relatively less complex structure of the endosperm due to its role as a storage tissue, compared to embryos or early seeds, which must differentiate into multiple organs.

The dynamic expression patterns and functional enrichment of TS genes indicated their involvement in specific tissues and stages of seed development (Supplementary Fig. 13b, c). For instance, genes specific to early whole seeds were primarily associated with cell wall biosynthesis and structural integrity, emphasizing the importance of creating space inside maternal tissues for rapid expansion of the embryo and endosperm. Embryo-specific genes were predominantly expressed in later stages of embryo development, indicating an early focus on general growth and tissue differentiation; whereas endosperm-specific genes were expressed throughout its development, emphasized its long-term programming (mediated through RNA processing and regulation) focused on starch and protein storage. Genes that were only expressed the embryo and/or endosperm (nowhere else in the plant) appeared to coordinate developmental processes and respond to environmental cues, particularly through sterol

biosynthesis and abscisic acid (ABA) responses. ABA in particular is a well-known player in seed maturation, dry-down, dormancy, germination, and both seed and whole-plant environmental responses[96].

While the functions of most seed-specific TFs remain unknown, our analysis revealed enrichment with known regulator families of seed development (e.g., WOX, NF-YB, NAC, ERF, AP2, MYB, Myb_related). These TF families are recognized for their key roles in events unique to seeds, especially in the formation and maturation of the endosperm and embryo[97–99]. This suggests that the remaining TS genes and TFs we identified may also hold regulatory roles in seed development. Future endeavors should focus on elucidating their roles by leveraging the genetic variation present in mutant and natural populations. The identified TS genes and TFs, along with the newly mapped gene networks governing starch and protein in BTx623, provide a crucial starting point for understanding how gene activity and metabolite accumulation is coordinated during seed development and organogenesis, as determined by SEM. These processes collectively influence the yield, quality, and nutrient profiles of sorghum grain.

## Materials and methods
### Plant material and field experiments
The study utilized the *Sorghum bicolor* cultivar 'BTx623,' cultivated under field conditions at the Quaker Research Farm in Lubbock, TX (33°35'52.9"N 101°54'21.4"W, elevation 992 m) during the summer of 2022. The farm experiences a semi-arid climate with an average yearly precipitation of 469 mm, primarily from May to October, and features Amarillo sandy clay loam soil[100]. Irrigation was maintained at 1 inch water per week.

Developing sorghum seeds were collected following successful pollination for detailed characterization (Supplementary Data 1). In brief, we conducted daily sampling from pollination until 30 dpa. Before flowering, panicles were covered with pollination bags to prevent cross-pollination. After pollination, these bags were replaced with mesh ones to safeguard seeds from birds while allowing light exposure and improved air circulation. From pollination to 15 dpa, we harvested the middle portion of 10 panicles that flowered on the same day for each replicate. Subsequently, for samples beyond 15 dpa, we collected the middle portion of 5 panicles per biological replicate. Three biological replicates for all data points were collected for transcriptome analysis and five replicates for 5, 10, 20, 25, and 30 dpa for metabolome analysis. Sampling was consistently conducted in the morning (between 9:00 AM to 11:00 AM) to minimize potential circadian influences. Following collection, samples were promptly transported to the laboratory, where they were dissected on ice using scalpels and tweezers to isolate embryos and endosperms.

For metabolome samples, 100 uniform seeds were isolated from the harvested panicles, flash-frozen in liquid nitrogen in 15 mL falcon tubes and stored at −80 °C. During seed dissection for embryo and endosperm RNA extraction, we isolated 50 uniform seeds from the harvested panicle. For early embryo samples (10–18 dpa), we isolated embryos from 40–50 seeds, whereas for later embryo samples, 20 seeds were sufficient for RNA extraction. Similarly, for endosperm samples (6–10 dpa), 40–50 seeds were used for isolation, while for later time points, 10 seeds were used for RNA extraction. Subsequently, embryo and endosperm sample were flash-frozen in liquid nitrogen and stored at −80 °C until further analysis.

### Kafirin analysis
Kafirin content analysis involved a step procedure to assess total kafirin levels and the degree of cross-linking (polymerization). Kafirin fractions were selectively extracted under non-reducing conditions (kafirin 1) and reducing conditions (kafirin 2) following the method outlined by Da Silva et al[30]. Seeds from days 5, 10, 15, 20, and 25 dpa were retrieved from −80 °C storage, immediately crushed using a mortar and pestle, and then returned to −80 °C. The coarsely crushed material was lyophilized and ground into a fine powder using a mortar and pestle. Kafirin 1 and kafirin 2 were then extracted as described in Da Silva, et al.[30]. except 50 mg of sample and 0.5 mL of solvent were used. Following extraction, beta-mercaptoethanol (BME) was added to kafirin 1 extracts to achieve a final volume of 2%. After

incubation with BME, samples underwent alkylation with 4-VP, as described in Bean, et al[101]. After kafirin 2 extraction, additional sample extraction solvent was added to equalize the total volume of kafirin 1 and kafirin 2. Kafirin 2 was then alkylated with 4-VP. Subsequently, kafirin 1 and kafirin 2 were subjected to analysis by RP-HPLC using C3 columns, following the procedure outlined in Bean, et al.[101].

### Metabolomic analysis
Untargeted metabolomic profiling was employed to analyze sorghum seeds using the LC-MS platform. The metabolites extraction and quantification were carried out by the service provider Innomics. For each replicate, whole seed were shipped in dry ice. For metabolite extraction, 50 mg of each sample was weighed into 1.5 mL Eppendorf tubes and immersed in a pre-cooled extraction solution (methanol: H2O = 7:3, v/v), supplemented with 20 μL of Internal Standard 1. Homogenization was conducted using a weaving grinder at 50 Hz for 10 minutes, followed by water bath ultra-sonication at 4 °C for 30 min. After being held at −20 °C for 1 h, the extracts were centrifuged at 14,000 rpm at 4 °C for 15 min. The resulting 600 μL supernatant was filtered using a 0.22 μm membrane, and 20 μL of the filtered solution from each sample was composited into the mixed quality control (QC) sample to assess the repeatability and stability of LC/MS analysis.

A Waters 2777c UPLC (Waters, USA) in series with a Q Exactive HF high-resolution mass spectrometer (Thermo Fisher Scientific, USA) was utilized for the separation and detection of metabolites. Post-experiment, the off-line mass spectrometry data were imported into Compound Discoverer v3.3 (Thermo Fisher Scientific, USA) software. Analysis of the mass spectrometry data, in conjunction with the BGI metabolome database (bmdb), mzCloud database (https://www.mzcloud.org/), and ChemSpider online database (https://www.chemspider.com/), resulted in a data matrix containing metabolite peak area and identification results. The identified metabolites were annotated using the Kyoto Encyclopedia of Genes and Genomes pathway (KEGG; https://www.genome.jp/kegg/) and Human Metabolome Databases (HMDB; https://hmdb.ca/)[102,103].A PCA and Partial Least Squares Discriminant Analysis (PLS-DA) were conducted with the metabolomics software MetaboAnalyst (https://www.metaboanalyst.ca/)[104]. Univariate analyses (*t*-tests) were used to calculate statistical significance (P-value). The following criteria were used to identify differentially expressed metabolites: Variable Importance in Projection value (VIP)> 1 and a P-value < 0.05, $\log_2$ (fold change) ≥ 1.5 or ≤ −1.5.

### RNAseq and data analysis
RiboPure™ RNA Purification Kit (AM1924; Invitrogen) was utilized for total RNA isolation following the manufacturer's instructions. We first used agarose gel electrophoresis, and a Nanodrop to assess the quality of the extracted RNA samples and check for DNA and protein contamination. At least 2 μg RNA was shipped to sequencing service provider Innomics on dry ice. RNA quality control and sequencing were performed by Innomics. Briefly, the RNA integrity number (RIN) value was calculated and samples with RIN ≥ 7 were used for RNA sequencing (Supplementary Data 5). Standard DNBSEQ Eukaryotic Transcriptome Resequencing protocols were followed for the construction of RNA-seq libraries, which were subsequently sequenced to generate PE150 reads by the service provider Innomics Inc. For the library construction, the fragmented mRNA was synthesized into first strand cDNA using random primers, while the second strand cDNA was synthesized with dUTP instead of dTTP. The synthesized cDNA was subjected to end-repair and 3' adenylated. Adaptors were ligated to the ends of these 3' adenylated cDNA fragments followed by the PCR amplifications. The raw data from the DNBSEQ platform was filtered to remove the adaptors, ployX and low-quality data by SOAPnuke software[105] with parameters: "-n 0.001 -l 20 -q 0.4 --adaMR 0.25 --ada_trim --polyX 50 --minReadLen 150".

To ensure data quality, the cleaned data underwent QC assessment using FastQC[106]. High-quality reads were aligned to the sorghum reference genome (version 3.3.1)[107,108] using STAR[109]. The FPKM values representing gene expression levels were calculated using StringTie[110].

Pearson correlation coefficients between biological replicates were calculated based on gene FPKM values, and replicates with a Pearson correlation > 0.8 were selected for further analysis.

Genes were considered expressed at a specific stage if they met the following criteria: (1) a minimum of two reads were mapped to the gene in each of two replicates, and (2) the average FPKM at a timepoint was ≥ 1 in at least one sample. To mitigate the impact of transcriptional noise, genes with a minimum FPKM value ≥ 1 in at least one sample were included for downstream analysis, consistent with the approach used in several other studies[34,35,46,99,111].

The PCA was employed to visually represent relationships among distinct seed tissue samples, utilizing the prcomp[112] function within R with default settings. Hierarchical clustering was performed by $k$-mean clustering with the pheatmap package[113] using default settings. The elbow method[114,115] was applied to determine the optimal cluster number. Transformed and normalized gene expression values with $\log_2$ (FPKM + 1) were used for PCA analysis and hierarchical clustering. For hierarchical clustering, relative expression values of the genes were calculated by dividing their expression level at different timepoints by their maximum observed RPKM.

Functional enrichment analysis was conducted based on a hypergeometric test using KEGG and ShinyGO (http://bioinformatics.sdstate.edu/go74/)[116]. Enriched KEGG pathways with an FDR < 0.05 were considered statistically significant, and selected KEGG pathways are presented. The co-expression network was generated through the STRING database (https://string-db.org/)[117]. Mean FPKM values were clustered using Fuzzy c-means clustering in the Mfuzz v2.42 R package (https://www.bioconductor.org/packages/release/bioc/html/Mfuzz.html)[118]. The optimal number of clusters was set to 12, and the fuzzifier coefficient was set to 2.01. Genes with a membership score of at least 0.5 were plotted and used as inputs for categorical enrichment analysis.

## qRT-PCR
The expression levels of selected genes were validated by quantitative real-time polymerase chain reaction (qRT-PCR). Total RNA was reverse transcribed into complementary DNA (cDNA) using iScript™ Reverse Transcription Supermix for RT-qPCR (Bio-Rad), according to the manufacturer's instructions. qRT-PCR was performed with two technical replicates for each of the three biological replicates using SsoAdvanced Universal SYBR Green Supermix (Bio-Rad) on a Bio-Rad CFX96 system. Data was processed using CFX Manager software. The relative transcript levels were normalized to the expression of the reference gene Serine/threonine-Protein Phosphatase (PP2A). It was selected according to the Sorghum reference genes selection paper[119]. Oligonucleotides used for these experiments are listed in Supplementary Data 14.

## Identification embryo, endosperm, and whole seed specific genes
A total of 42 previously published non-seed sorghum RNA-seq datasets were employed to identify the tissue-specific (TS) genes[67,108,120–127]. The TS genes were identified utilizing a TS scoring algorithm[128,129] that compares the expression level of a gene in a given compartment with its maximal expression level in the other sample. Therefore, TS scores range from 0 to 1, and the higher the TS score of a gene for a tissue, the more likely the gene is specifically expressed in that tissue[95,128,129]. This study defined TS genes as having a TS score > 0.5, following similar criteria used in other studies in maize[95] and yam[111].

## Statistics and reproducibility
Details of the statistical tests used in the study are provided in the respective methods sections and Supplementary Data. The RNA-seq, metabolome profiling and qPCR were performed with two, five and three independent biological replicates, respectively. Pearson correlations (r) among the replicates were calculated based on the expression levels (FPKM) of the genes. The assessment of GO term enrichment was conducted using Fisher's Exact test, followed by adjustment for false discovery rate (FDR) as implemented in

PANTHER19.0. For metabolome pathway enrichment analysis, the Hypergeometric Test was utilized, as implemented in MetaboAnalyst.

## Reporting summary
Further information on research design is available in the Nature Portfolio Reporting Summary linked to this article.

## Data availability
The source data behind the graphs in the main figures can be found in Supplementary Data 15. The RNA-seq data has been deposited into the Ensembl ArrayExpress collection in BioStudies under Accession number: E-MTAB-13406. The metabolomic data, encompassing compound names, formulas, exact Q1 (m/z) values, molecular weights, and peak intensities, are available in the supplementary data.

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

## Acknowledgements

This research was supported by the intramural research program of the U.S. Department of Agriculture, National Institute of Food and Agriculture, Agriculture and Food Research Initiative (AFRI), under the award number: 2023-67013-39631. Y.J. and A.K. was also supported by the State of Texas' Governor's University Research Initiative (GURI). S.R.B. was funded by USDA-ARS project number 3020-43440-002.The findings and conclusions in this preliminary publication have not been formally disseminated by the U. S. Department of Agriculture and Should not be construed to represent any agency determination or policy. Names are necessary to report factually on available data; however, the U.S. Department of Agriculture neither guarantees nor warrants the standard of the product and use of the name by the U.S. Department of Agriculture implies no approval of the product to the exclusion of others that may also be suitable. USDA is an equal opportunity provider and employer.

## Author contributions

Y.J. conceived of and designed the project. A.K. and R.T. performed the experiments and analyzed the data. S.B. conducted protein analysis. A.K., Y.J., R.T. and M.K.Y. prepared the manuscript. All authors edited and approved the final version for publication.

## Competing interests

The authors declare no competing interests.
