## [Peer Review File · Communications Biology]

Reviewers' comments:

Reviewer #1 (Remarks to the Author):

This manuscript carried out transcriptomic and metabolomic analyses of developing seeds, embryos and endosperm in sorghum, and identified some genes and pathways related to starch and protein synthesis. The followings were suggested to be considered in revising the manuscript.

- (i) According to the formation of molecules, it is suggested to move the transcriptomic analyses part ahead that of metabolomic analyses. Moreover, at least three biological replicates should be used in transcriptomic analyses, but only two replicates were used in the present study.
- (ii) Procedure cell death usually take place in the development of sorghum endosperm, and this process should be considered in analyzing the gene expression in sorghum endosperm.
- (iii) In 'Hub genes and key networks associated with starch and protein synthesis' section, the genes related to protein (such as kafirin) synthesis were analyzed, and the authors should pay more attention to the analysis of starch synthesis genes.
- (iv) The authors summarized some results in the Discussion part, while some specific scientific problems should be focused in this part.
- (v) As to REFERENCES list, the authors should pay more attention to the writing of journal names, scientific names of plants in the cited papers.

Reviewer #2 (Remarks to the Author):

Authors of the manuscript no COMMSBIO-23-5012 have analysed spatiotemporal transcriptome and metabolome profiles throughout embryo and endosperm development in the sorghum reference genome line, BTx623.

Regarding the experimental part of work, the authors have used advanced and suitable methodology. The experiments are well designed and carefully performed. The amount of data generated by the work is impressive; authors have sequenced 45 different transcriptomic libraries and performed profiling of 5 different metabolomes (according to the Table S1). The information included in tables and figures is rather clear (apart from issues commented below). The results are appropriately discussed; the importance and novelty of presented work is justified; hypotheses are drawn and the potential for the industrial application is highlighted. The manuscript is written with adequate English.

The work has a potential to bring novel information to the field of study, however, there are several issues which should be addressed or explained before considering this work for publishing in Communications Biology.

1. Materials and methods; Plant material and field experiments section needs to be improved with detailed description of sampling for RNA-seq and metabolomic analysis. Please, move the sampling description from the Results section to Materials and methods and refer to the Table S1. Please, also state more clearly how many seed replicates were harvested for each analysis and how

many technical replicates of RNA-seq or LC-MS analysis were made. I would suggest to better navigate the reader throughout the manuscript.

2. Materials and methods; Metabolomic analysis. Please, provide links and/or references to the metabolic databases you used.

3. Materials and methods; RNAseq and data analysis. Agarose gel electrophoresis is definitely not considered sufficient to assess the integrity of the RNA, especially for such downstream applications as RNA-seq. Have you estimated the RNA integrity number (RIN) for your RNA samples and was it high enough for the library construction and RNAseq?

4. Materials and methods; RNAseq and data analysis. Please, provide more details regarding the cDNA library construction. What was the name of the Illumina kit for the library synthesis/ oligo (dT) beads/adapters - add names of the products and names of manufacturers. Provide details of the Illumina platform. Have you performed the RNA sequencing "in house" or was it done commercially, if so, provide the company details. Remember that your experiment should be reproducible for the potential reader.

5. Data availability. The reader has no access to the link provided. When the ID E-MTAB-13406 was searched for, it did not match any studies. Can you explain this?

6. Figure 1c. Please, correct the colouring of the bars. Also, how is it possible, that at 20 dpa the kafirin 1 content exceeded the total protein content?

7. Table S2 is missing the table description.

8. Results. There are sentences, which should rather be moved to the Discussion part, such as in lines: 201, 224-225, 241-243, 284-285, 299-301, 341-348. Please revise your text such that the Results section only treats the raw results and reserve all your interpretations for the Discussion section.

9. Is there any particular reason, why the gene expression results from RNA-seq were not confirmed by qRT-PCR for at least some of the target genes?

10. The analysis of such a massive RNAseq data set is impressive with the clustering method and co-expression network, but the final outcome does not feel satisfactory enough for the reader. You selected a number of genes which you designate as hub genes following different search criteria, you have listed them in several tables (Table S6 – S11) and presented in several figures (Figure 4 and 5, Figure S6 – S11) but you do not name them. The gene ID is not informative for the reader, selected hub genes are missing gene descriptions (Table S6 – S11), which would indicate their function. Crosschecking with the Data S2 table for each single gene is not convenient for the potential user. Additionally, I would suggest to provide an additional Figure with kafirin and starch (simplified?) biosynthetic pathways, where the most important hub/candidate genes would be highlighted. In the current form the huge data set you created feels underutilised, which might be just due to the way of presentation.

Summarising, the manuscript should be improved, as suggested and the issues should be addressed or explained. After the improvements are done, I would recommend presented manuscript for reconsidering.

Reviewer #3 (Remarks to the Author):

Understanding the molecular processes governing the coordinated development of the embryo and endosperm is crucial for enhancing the yield and nutritional quality of cereal grains. In this study, the authors analyzed spatiotemporal transcriptome and metabolome profiles throughout embryo and endosperm development in the sorghum reference genome line, BTx623. From the perspective of sorghum functional genome research, this work provides a very meaningful resource. However, I have a few of major concerns that need the authors to address:

1. In ABSTRACT, the authors do not summarize their findings, but emphasize the importance of their own work, which is not in line with the writing form of the abstract;
2. Dynamic metabolic changes during sorghum seed development : In the summary of this part of the results, the authors say that, line198-201 “THESE FINDINGS SUGGESTED THAT THE STARCH AND FATTY ACID CONTENTS IN SORGHUM SEEDS WERE DETERMINED BEFORE FINAL PROTEIN CONTENT, POTENTIALLY CONTRIBUTING TO THE WELL-KNOWN NEGATIVE CORRELATION BETWEEN STARCH AND PROTEIN CONTENT. SIMILAR PATTERNS HAVE BEEN REPORTED IN OTHER CROPS SUCH AS PEAS AND MAIZE 64-66.” however, for the metabolome analysis of sorghum, what are the specific findings of the work? The authors did not try to give an answer.
3. The transcriptome landscape of sorghum seed development: In the part of transcriptome analysis, the authors carried out a classical co-expression network analysis and obtained co-expression modules and hub genes. However, I don't think anything like this can be called a meaningful discovery or conclusion: line299-line300: THIS DIVERGENT PROTEIN EXPRESSION PATTERN BETWEEN EMBRYO AND ENDOSPERM TISSUES UNDERSCORES THE PIVOTAL ROLE OF ENDOSPERM DEVELOPMENT IN DETERMINING THE FINAL PROTEIN CONTENT OF SORGHUM SEEDS IN BTX623.
4. To sum up, the author found a good experimental Angle, but did not dig out the unique place in the relationship between development and metabolism during the development of sorghum kernel, that is, did not find out the key scientific problems, so the findings of this study are insufficient, and can only be a research resource.

Point-to-point Response

Reviewer #1 (Remarks to the Author):

This manuscript carried out transcriptomic and metabolomic analyses of developing seeds, embryos and endosperm in sorghum, and identified some genes and pathways related to starch and protein synthesis. The followings were suggested to be considered in revising the manuscript.

(i) According to the formation of molecules, it is suggested to move the transcriptomic analyses part ahead that of metabolomic analyses.

Response: Thank you for the suggestions. We integrated metabolomic analysis alongside SEM imaging and kafirin analysis to provide as the phenotypic characterization of sorghum seed development. Additionally, we acknowledge the importance of further exploring gene expression analysis to elucidate the observed phenotype. To enhance clarity, we have subdivided this section into two distinct subtitles: "Morphological Analyses of Sorghum Seed Development" and "Dynamic Metabolic Changes During Sorghum Seed Development."

Moreover, at least three biological replicates should be used in transcriptomic analyses, but only two replicates were used in the present study.

Response: We appreciate your concern regarding the addition of more replicates. However, due to the extensive labor and budget constraints associated with our large-scale transcriptome project, we cannot add additional replicates at this time. Nonetheless, we conducted rigorous correlation analysis between the two replicates, ensuring their agreement with each other. The robust correlation, evidenced by an average R^2 of 0.976 (Supplementary data 5, Line 244-245 in the main manuscript), validates the high quality of our data. In response to Reviewer #2's suggestion and following discussions with the editor, we have performed qRT-PCR to further validate the RNAseq data. In brief, the qPCR results from three replicates of four randomly selected genes at the five major timepoints (5, 10, 15, 20, 25 dpa) closely matched the RNAseq data (Supplementary Fig. 5), further validating the reliability of our findings (line 245-248).

(ii) Procedure cell death usually take place in the development of sorghum endosperm, and this process should be considered in analyzing the gene expression in sorghum endosperm.

Response: We agree programmed cell death (PCD) is programmed cell death (PCD) is a crucial cellular process in endosperm development. In this revision, we have incorporated an analysis of gene expression of the Ethylene biosynthesis and four previously identified regulators of PCD in maize endosperm development (see lines 312-323 and Supplementary Fig. 8). In summary, our findings reveal a down regulation trend in these gene groups throughout sorghum endosperm development.

(iii) In 'Hub genes and key networks associated with starch and protein synthesis' section, the genes related to protein (such as kafirin) synthesis were analyzed, and the authors should pay more attention to the analysis of starch synthesis genes.

Response: Thank you for pointing this out. We have reorganized the initial segment of this section to describe the expression patterns of established starch synthesis enzymes alongside kafirin coding genes. Particularly, emphasis has been placed on the notable expression levels of

kafirin coding genes within the endosperm. Subsequently, the latter portion focuses on network analysis findings, shedding light on potential regulatory mechanisms governing starch and protein accumulation in sorghum seeds. See the change from line 331 to 353.

(iv) The authors summarized some results in the Discussion part, while some specific scientific problems should be focused in this part.

Response: In response to the reviewer's suggestion, we have reorganized the discussion section accordingly. The initial segment focuses on elucidating the diversity in seed development within the sorghum population. We underscore how our examination of seed development in the reference genome line establishes a crucial foundation for comprehending the extensive diversity present in sorghum seed traits. In the subsequent portion, we delve into seed development-specific genes. It's noteworthy that a significant proportion of the seed-specific genes we uncovered lack functional characterization. Drawing insights from research conducted on other crops, we explore the potential significance of these genes in facilitating seed development. Please see the revision in the new Discussion part.

(v) As to REFERENCES list, the authors should pay more attention to the writing of journal names, scientific names of plants in the cited papers.

Response: Thank you for the suggestion. We have confirmed the formatting of the reference list aligning with the requirements of the Communications Biology journal.

Reviewer #2 (Remarks to the Author):

Authors of the manuscript no COMMSBIO-23-5012 have analysed spatiotemporal transcriptome and metabolome profiles throughout embryo and endosperm development in the sorghum reference genome line, BTx623.

Regarding the experimental part of work, the authors have used advanced and suitable methodology. The experiments are well designed and carefully performed. The amount of data generated by the work is impressive; authors have sequenced 45 different transcriptomic libraries and performed profiling of 5 different metabolomes (according to the Table S1).

The information included in tables and figures is rather clear (apart from issues commented below). The results are appropriately discussed; the importance and novelty of presented work is justified; hypotheses are drawn and the potential for the industrial application is highlighted. The manuscript is written with adequate English.

Response: Thank you for the positive evaluations of our work.

The work has a potential to bring novel information to the field of study, however, there are several issues which should be addressed or explained before considering this work for publishing in Communications Biology.

1. Materials and methods; Plant material and field experiments section needs to be improved with detailed description of sampling for RNA-seq and metabolomic analysis. Please, move the sampling description from the Results section to Materials and methods and refer to the Table S1. Please, also state more clearly how many seed replicates were harvested for each analysis

and how many technical replicates of RNA-seq or LC-MS analysis were made. I would suggest to better navigate the reader throughout the manuscript.

Response: Thank you for the comments. We have carefully revised the methodology section of the manuscript, incorporating detailed descriptions of the sampling procedures for RNA-seq and metabolomic analysis as suggested. Table 1 (new number is Supplementary Data 1) has been referred in the method as well. The first paragraph is about the morphological characterization of the seed development including the change of color and weight. The subtitle was changed to make it clear.

Additionally, we have provided clearer statements regarding the number of seed replicates harvested for each analysis and the number of technical replicates conducted for RNA-seq or LC-MS analysis. Please refer to lines 503-525 for the revised methodology section. We believe these updates enhance the clarity and completeness of the methodology, improving the overall navigability of the manuscript.

2. Materials and methods; Metabolomic analysis. Please, provide links and/or references to the metabolic databases you used.

Response: Thank you for the suggestion. We have now included references and hyperlinks to all the databases accessed. These can be found in the revised Materials and Methods section 'Metabolomic Analysis'. Line 594, 597, 598, 600. Line 645, 648, 650 under the section of RNAseq and data analysis for RNAseq related databases.

3. Materials and methods; RNAseq and data analysis. Agarose gel electrophoresis is definitely not considered sufficient to assess the integrity of the RNA, especially for such downstream applications as RNA-seq. Have you estimated the RNA integrity number (RIN) for your RNA samples and was it high enough for the library construction and RNAseq?

Response: Yes, the RNA integrity number (RIN) was evaluated for all RNA samples, and they have sufficiently high values for library construction. Detailed information regarding our RNA integrity assessment procedures is provided in the methods section, specifically outlined in lines 607-624. To briefly summarize, agarose gel electrophoresis was initially utilized to assess the quality of the RNA samples. Subsequently, the RNA integrity of all samples was evaluated using the RNA integrity number (RIN) measurement conducted on the bioanalyzer at the sequencing facility. Notably, all samples displayed excellent RIN values (≥ 7), thereby reinforcing their suitability for library construction and subsequent RNA-seq analysis (Supplemental Data 5).

4. Materials and methods; RNAseq and data analysis. Please, provide more details regarding the cDNA library construction. What was the name of the Illumina kit for the library synthesis/ oligo (dT) beads/adapters - add names of the products and names of manufacturers. Provide details of the Illumina platform. Have you performed the RNA sequencing "in house" or was it done commercially, if so, provide the company details. Remember that your experiment should be reproducible for the potential reader.

Response: Thank you for the inquiry. The library construction and sequencing was carried out following the DNBSEQ Eukaryotic Transcriptome Resequencing platform protocols by the commercial service provider Innomics Inc. More details were added to the method section line 615-624.

5. Data availability. The reader has no access to the link provided. When the ID E-MTAB-13406 was searched for, it did not match any studies. Can you explain this?

Response: As detailed in the manuscript, our data has been deposited in Ensembl ArrayExpress under the accession number E-MTAB-13406. This dataset will be made publicly available following the acceptance of our manuscript for publication. This explains why direct searches using the ID may not yield results. For early review, you can access the data through the following link: <https://www.ebi.ac.uk/biostudies/arrayexpress/studies/E-MTAB-13406?key=b4d13b86-036a-491a-a4da-481faa48c2ff>

We have also attached a screenshot for your reference.

BioStudies. Search ArrayExpress
Examples: E-MEXP-31, cancer

Home | BioStudies | ArrayExpress | Submit | Help | About BioStudies | Feedback | Login

ArrayExpress Functional genomics data

BIOSTUDIES / ARRAYEXPRESS / E-MTAB-13406

Release Date: 12 July 2024 • Modified: 9 October 2023 • Private Share

Spatiotemporal Transcriptomics and Metabolomic Analysis Provide Insights into the Nutrition Synthesis in Developing Sorghum Seed

Accession E-MTAB-13406

Study type RNA-seq of coding RNA EF0

Organism Sorghum bicolor

Description In this study, we developed a high-resolution spatiotemporal gene expression profiles for sorghum embryo and endosperm, from fertilization through maturation. we identified genes and transcription factors strongly associated with starch and kafirins synthesis. In addition, we pinpointed 499 genes exclusively expressed in seeds, among which 41 were transcription factors, playing pivotal roles in advancing the seed development process.

Protocols show table

Name	Size	Section	Desc
pricessed_matrix.txt	12.5 MB	Processed Data	Proc Date
E-MTAB-13406.idf.txt	11 KB	MAGE-TAB Files	Inve Desi Forn Sam

6. Figure 1c. Please, correct the colouring of the bars. Also, how is it possible, that at 20 dpa the kafirin 1 content exceeded the total protein content?

Response: Thank you for bringing this to our attention. Regarding the coloring of the bars in Figure 1c, we have corrected the color scheme to ensure clarity and accuracy in representing the data. As for your question about the kafirin 1 content exceeding the total protein content at 20 dpa, we acknowledge the discrepancy. The initial observation of kafirin 1 content exceeding total protein content at 20 dpa was based on data from only one replicate. Unfortunately, during quantification, we encountered technical issues that led to the loss of two replicates for this sample. After the submission of the paper, we repeated the sampling and quantification procedures for the 20 dpa sample. The updated figure now reflects data based on three replicates, providing a more reliable representation of the kafirin 1 content relative to the total protein content at 20 dpa. We appreciate your attention to detail and apologize for any confusion caused by the initial oversight.

7. Table S2 is missing the table description.

Response: Thank you for bringing this to our attention. We have now included the description for Table S2, ensuring that readers have a clear understanding of the table content and its relevance to the study.

8. Results. There are sentences, which should rather be moved to the Discussion part, such as in lines: 201, 224-225, 241-243, 284-285, 299-301, 341-348. Please revise your text such that the Results section only treats the raw results and reserve all your interpretations for the Discussion section.

Response: Thank you for the suggestion. We have removed the comparison with maize in the lines the reviewer mentioned: 201, 224-225, 299-301. The other three mentioned above were kept as the interpretation of the result. We also have carefully screened the whole manuscript to remove any similar discussion sentences.

9. Is there any particular reason, why the gene expression results from RNA-seq were not confirmed by qRT-PCR for at least some of the target genes?

Response: Following this comment and discussions with the editor, we have performed qRT-PCR to further validate the RNA-seq data. In brief, the qPCR results from three replicates of four randomly selected genes at the five major timepoints (5, 10, 15, 20, 25 dpa) closely matched the RNAseq data (Supplementary Fig. 5), further validating the reliability of our data (line 245-248).

10. The analysis of such a massive RNAseq data set is impressive with the clustering method and co-expression network, but the final outcome does not feel satisfactory enough for the reader. You selected a number of genes which you designate as hub genes following different search criteria, you have listed them in several tables (Table S6 – S11) and presented in several figures (Figure 4 and 5, Figure S6 – S11) but you do not name them. The gene ID is not informative for the reader, selected hub genes are missing gene descriptions (Table S6 – S11), which would indicate their function. Crosschecking with the Data S2 table for each single gene is not convenient for the potential user.

Response: We appreciate your feedback regarding the accessibility of the hub gene information. In response, we have made revisions to Tables S6-S11. These revisions include the addition of gene names alongside their corresponding orthologs in rice and Arabidopsis, as well as brief descriptions of their functions.

Additionally, I would suggest to provide an additional Figure with kafirin and starch (simplified?) biosynthetic pathways, where the most important hub/candidate genes would be highlighted. In the current form the huge data set you created feels underutilised, which might be just due to the way of presentation.

Response: Thank you for your insightful suggestion. After careful consideration, we have decided not to include biosynthetic pathway figures in the current paper. Accurately representing all the hub genes associated with starch and kafirin biosynthesis within a single figure without compromising clarity seems difficult. However, we have focused on enhancing the accessibility and clarity of the hub gene information by providing detailed supplementary data. These tables now include gene names along with their orthologous information in rice and Arabidopsis, as well as brief descriptions of their functions, specifically for the hub genes related to starch and kafirin.

Summarising, the manuscript should be improved, as suggested and the issues should be addressed or explained. After the improvements are done, I would recommend presented manuscript for reconsidering.

Reviewer #3 (Remarks to the Author):

Understanding the molecular processes governing the coordinated development of the embryo and endosperm is crucial for enhancing the yield and nutritional quality of cereal grains. In this study, the authors analyzed spatiotemporal transcriptome and metabolome profiles throughout embryo and endosperm development in the sorghum reference genome line, BTx623. From the perspective of sorghum functional genome research, this work provides a very meaningful resource. However, I have a few of major concerns that need the authors to address:

1. In ABSTRACT, the authors do not summarize their findings, but emphasize the importance of their own work, which is not in line with the writing form of the abstract;

Response: Thank you for your valuable suggestion. We have carefully revised the abstract to incorporate the major findings, elucidating the pathways and highlighting key hub genes. Additionally, we have ensured that the formatting adheres to the journal's specifications, encompassing a concise 150-word limit. We believe these enhancements have strengthened the clarity and accessibility of our work, aligning it more closely with the journal's guidelines.

2. Dynamic metabolic changes during sorghum seed development : In the summary of this part of the results, the authors say that, line198-201 “THESE FINDINGS SUGGESTED THAT THE STARCH AND FATTY ACID CONTENTS IN SORGHUM SEEDS WERE DETERMINED BEFORE FINAL PROTEIN CONTENT, POTENTIALLY CONTRIBUTING TO THE WELL-KNOWN NEGATIVE CORRELATION BETWEEN STARCH AND PROTEIN CONTENT. SIMILAR PATTERNS HAVE BEEN REPORTED IN OTHER CROPS SUCH AS PEAS AND MAIZE 64-66.” however, for the metabolome analysis of sorghum, what are the specific findings of the work? The authors did not try to give an answer.

Response: Thank you for bringing this to our attention. We have revised the text for clarity, specifically addressing the concern mentioned in Line 223-226. The conclusion regarding the earlier accumulation of starch than protein is derived from the enriched metabolites observed at various timepoints.

Our metabolite profiling analysis is structured in two main parts: Firstly, we presented the analysis of 2073 detected compounds during sorghum seed development and their pathway enrichment at different timepoints, as depicted in Supplementary Fig 2 and detailed in the first paragraph of this section. Subsequently, we delved into a comparison of the differential accumulation of compounds between different timepoints, revealing the dynamic changes in metabolite profiles throughout seed development. This is elucidated in the second paragraph of the section, where we also highlighted the 189 and 234 consistently up-regulated and down-regulated metabolites, shedding light on their potential roles in seed development."

3. The transcriptome landscape of sorghum seed development: In the part of transcriptome analysis, the authors carried out a classical co-expression network analysis and obtained co-expression modules and hub genes. However, I don't think anything like this can be called a meaningful discovery or conclusion: line299-line300: THIS DIVERGENT PROTEIN EXPRESSION PATTERN BETWEEN EMBRYO AND ENDOSPERM TISSUES UNDERSCORES THE PIVOTAL ROLE OF ENDOSPERM DEVELOPMENT IN DETERMINING THE FINAL PROTEIN CONTENT OF SORGHUM SEEDS IN BTX623.

Response: As also pointed out by another reviewer, we acknowledge that this discussion was not accurate. In response, we have removed the problematic sentence and rephrased the first section of the discussion for improved accuracy and clarity.

4. To sum up, the author found a good experimental Angle, but did not dig out the unique place in the relationship between development and metabolism during the development of sorghum kernel, that is, did not find out the key scientific problems, so the findings of this study are insufficient, and can only be a research resource.

Response: The primary objective of our project is to elucidate the molecular landscape of sorghum seed development, aiming to provide the sorghum community with a foundational resource for understanding gene functions crucial to this important developmental process. A key implication of releasing this resource is the identification of target genes with potential to enhance the sorghum grain quality. Notably, we are currently engaged in characterizing hub genes involved in starch and protein synthesis as part of our subsequent projects. We understand your concern. In response to the reviewers' and editors' suggestions, we have made substantial revisions to the manuscript to address these valuable insights."

REVIEWERS' COMMENTS:

Reviewer #1 (Remarks to the Author):

The authors have revised the manuscript according to the reviewers' suggestions, but further minor revisions are necessary. (i) Figure 2 cannot be seen in the revised manuscript. (ii) Please check the writing of plant scientific names in the Reference list, such as No. 10, 11, 13, 25, 43, 48, 52, 53, 54, 56, 58 etc., and check the writing of journal names in the Reference list, such as No. 26, 30, 31, 35, 37, 39, 41, 45, 54, 57, 61 etc.

Reviewer #2 (Remarks to the Author):

Dear Editor,

Dear Authors,

Compared to its previous version, I find the current form of the manuscript to be significantly improved. Indeed, all comments regarding the original manuscript were addressed and appropriate changes were made. Therefore, I recommend the manuscript in its current form for publication.